# Ultrafast water sensing and thermal imaging by a metal-organic framework with switchable luminescence

Ling Chen[1], Jia-Wen Ye[1], Hai-Ping Wang[1], Mei Pan[1], Shao-Yun Yin[1], Zhang-Wen Wei[1], Lu-Yin Zhang[1], Kai Wu[1], Ya-Nan Fan[1] & Cheng-Yong Su[1,2]

A convenient, fast and selective water analysis method is highly desirable in industrial and detection processes. Here a robust microporous Zn-MOF (metal–organic framework, Zn(hpi2cf)(DMF)(H$_2$O)) is assembled from a dual-emissive H$_2$hpi2cf (5-(2-(5-fluoro-2-hydroxyphenyl)-4,5-bis(4-fluorophenyl)-1$H$-imidazol-1-yl)isophthalic acid) ligand that exhibits characteristic excited state intramolecular proton transfer (ESIPT). This Zn-MOF contains amphipathic micropores ($<$3 Å) and undergoes extremely facile single-crystal-to-single-crystal transformation driven by reversible removal/uptake of coordinating water molecules simply stimulated by dry gas blowing or gentle heating at 70 °C, manifesting an excellent example of dynamic reversible coordination behaviour. The interconversion between the hydrated and dehydrated phases can turn the ligand ESIPT process on or off, resulting in sensitive two-colour photoluminescence switching over cycles. Therefore, this Zn-MOF represents an excellent PL water-sensing material, showing a fast (on the order of seconds) and highly selective response to water on a molecular level. Furthermore, paper or *in situ* grown ZnO-based sensing films have been fabricated and applied in humidity sensing (RH$<$1%), detection of traces of water ($<$0.05% v/v) in various organic solvents, thermal imaging and as a thermometer.

[1] MOE Laboratory of Bioinorganic and Synthetic Chemistry, State Key Laboratory of Optoelectronic Materials and Technologies, Lehn Institute of Functional Materials, School of Chemistry, Sun Yat-Sen University, Guangzhou 510275, China. [2] State Key Laboratory of Applied Organic Chemistry, Lanzhou University, Lanzhou 730000, China. Correspondence and requests for materials should be addressed to M.P. (email: panm@mail.sysu.edu.cn) or to C.-Y.S. (email: cesscy@mail.sysu.edu.cn).

Metal–organic frameworks (MOFs) have triggered enormous interest for potential application in smart and efficient photoluminescence (PL) sensing and imaging in recent years[1–12]. Because of their hybrid porous nature, MOFs can provide feasible in-and-out channels for guest molecules and behave like a sponge breathing under the driving forces of chemicals, light, electricity, pressure, temperature and so on[13–18]. This is then reflected in the emission change of the host and makes it possible for guest detection. However, the selectivity of a certain target molecule (gas or liquid) by a porous PL sensor is often affected by other molecules with similar or smaller size and similar properties, which may also enter the pore and change the emission analogously[4,15,19,20]. For example, oxygen-sensing MOF sensors based on triplet-state oxygen quenching always show similar luminescence quenching when exposed to other molecules with triplet electronic ground state such as nitric oxide and nitrobenzene[21]. Therefore, design of a highly selective MOF-based PL sensor featuring special attributes of the MOF structure for unique interactions with a certain kind of guest molecules is still challenging[9,22–24].

Water sensing and detecting in gases or organic products is of great importance in chemical industrial processes, environmental monitoring, pharmaceutical and food inspection and so on[25–28]. Compared with the traditional Karl Fischer titration and electrochemical methods, the PL water sensing based on fluorescent organic molecules or metal–organic hybrid materials represents an advantageous choice due to various versatilities, such as easy preparation, simple operability, high sensitivity, convenient on-site and non-invasive detection and so on[29]. Especially, a few fast (on the order of minutes) and highly sensitive (<1% v/v) water PL sensors have been accomplished based on some flexible and luminescent MOFs[22,30–33], indicating that development of fast and practical water PL sensors are expected by more delicate design of MOF materials with adequate porosity and switchable luminescence behaviour.

Here we report assembly of a Zn-MOF (LIFM-CL1) as a unique water sensor from a purposely designed $H_2hpi2cf$ ligand with dual-emission behaviour originated in excited state intramolecular proton transfer (ESIPT)[34–39]. The as-prepared hydrated LIFM-CL1-$H_2O$ [Zn(hpi2cf)(DMF)($H_2O$), DMF = N,N-dimethylformamide, $H_2hpi2cf$ = 5-(2-(5-fluoro-2-hydroxyphenyl)-4,5-bis(4-fluorophenyl)-1H-imidazol-1-yl)isophthalic acid] features amphipathic microporosity (<3 Å) with surprisingly facile and reversible removal/uptake of coordinating water molecules under mild conditions (dry gas blowing or gentle heating)[40–42], giving rise to the dehydrated form LIFM-CL1 [Zn(hpi2cf)(DMF)] via a single-crystal-to-single-crystal (SC–SC) transformation (LIFM-CL1-$H_2O$ ↔ LIFM-CL1 + $H_2O$) and simultaneously ESIPT-related two-colour luminescence switching[43,44]. This swift water-driven structural transformation and PL-switching property makes the present Zn-MOF one of the most powerful water sensors known so far owing to its hypersensitive response towards water molecules on a molecular level[45], showing the following characteristics: the fastest PL responsiveness on the order of seconds, solely water interaction, easy stimulation and nondestructive regeneration, convenient material utilization from either hydrated or dehydrated forms, and multiply detection ways including emission wavelength, intensity or Commission Internationale de L'Eclairage (CIE) coordinate. Therefore, ultrafast and highly sensitive real-time and on-site detection of a trace of water (<0.05% v/v) in various organic solvents and gas humidity (relative humidity (RH)<1%) is achievable. Furthermore, this microporous Zn-MOF can be fabricated into paper-based or in situ ZnO precursor-supported hybrid films for convenient water sensing and thermal imaging for practical purposes.

## Results

**Structural transformation and sensing mechanism.** The hydrated LIFM-CL1-$H_2O$ with the formula of [Zn(hpi2cf)(DMF)($H_2O$)] was obtained from the solvothermal reaction of ESIPT-ligand $H_2hpi2cf$ with $Zn(NO_3)_2 \cdot 6H_2O$, which is insoluble in water and other common organic solvents. The single-crystal structural analyses reveal that each $Zn^{2+}$ connects two different hpi2cf$^{2-}$ ligands through carboxylate O-atoms, and further coordinates with one DMF and one water molecules (Fig. 1). Two carboxylate groups of each ligand join two different Zn centres, thus generating one-dimensional chains (Supplementary Fig. 2), which are aligned in parallel along b axis and consolidated into three-dimensional framework via formation of interchain hydrogen bonds (HBs) and π-stacking. Interestingly, the in situ X-ray diffraction tests of the same crystal under the blow of dry $N_2$ at room temperature (25 °C) or moderate heating disclose a fast reversible SC–SC transformation with slightly shrunk unit cell (Supplementary Table 1). The structural conversion is caused by escape of coordination water to lead to dehydrated [Zn(hpi2cf)(DMF)] (LIFM-CL1). Structural analyses unveil that water removal is facilitated by formation of Zn–O bond between closely adjacent Zn1 and O2 belonging to two neighbouring chains (Fig. 1), thus cross-linking one-dimensional chains into two-dimensional layer on bc-plane (Supplementary Fig. 2). This structural transformation only involves local Zn-coordination change, leaving the overall crystal packing almost intact, and represents a rather unique example showing dynamic reversible coordination behaviour.

The H-bonding information is obtained from analysis of the hydrated LIFM-CL1-$H_2O$ single-crystal at low temperature of 150 K. As seen from Fig. 2 and Supplementary Fig. 2, the coordinating water molecule (O6) is located in a micropore (<3 Å) and forms HBs with the uncoordinating carboxylate

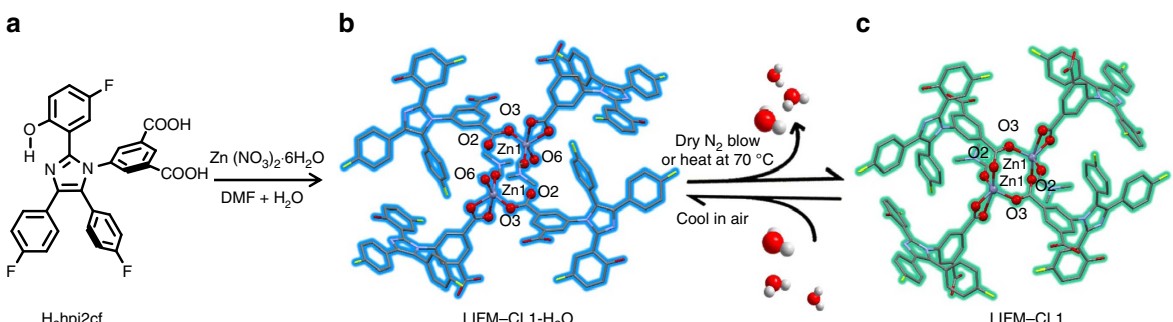

**Figure 1 | Synthetic route and structural transformation.** (**a**) Ligand, (**b**) hydrated LIFM-CL1-$H_2O$ (blue) and (**c**) dehydrated LIFM-CL1 (cyan), showing coordination environmental change around Zn centres. All hydrogen atoms are omitted for clarity.

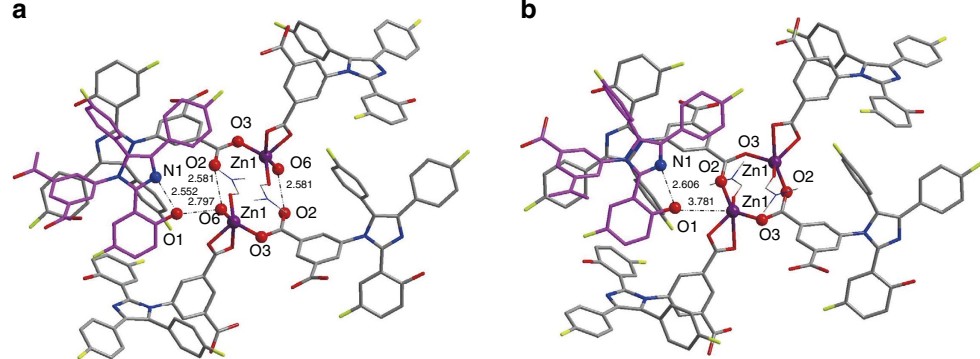

**Figure 2 | Crystal structures.** Local structural comparison between hydrated LIFM-CL1-H$_2$O (**a**) and dehydrated LIFM-CL1 (**b**), showing H-bonding variation relating to the hydroxyl –OH group and imine N-atom along structural transformation.

O2-atom of a neighbouring chain and the hydroxyl O1-atom of another neighbouring chain (O···O, 2.581 and 2.797 Å). Meanwhile, the –OH group also forms intramolecular HB with the imine N-atom (O···N, 2.552 Å). In contrast, on removal of a coordinating water molecule (O6) to transfer to the dehydrated LIFM-CL1, Zn1–O2 binding occurs and leaves the hydroxyl O1-atom of another neighbouring chain far away (O···O separation ≥3.781 Å) without any intermolecular HBs, only preserving the intramolecular O···N HB (O···N, 2.606 Å) between the –OH group and the imine N-atom.

From the structural determination, the hpi2cf$^{2-}$ ligands in both LIFM-CL1-H$_2$O and LIFM-CL1 are in enol form. However, the intermolecular H-bonding difference imparts distinct effect on the ESIPT processes in these two interconvertible structures[39]. It is known that the ESIPT process brings forward unique dual emission in organic molecules by the fast switching between enol (*E*) and keto (*K*) tautomers (Supplementary Fig. 3), which is subject to subtle surrounding environmental disturbance on the intramolecular proton transfer and leads to tunable emission for PL sensing[45–52]. As demonstrated in Fig. 3 and Supplementary Fig. 3, the H-bonding between water and –OH group in hydrated LIFM-CL1-H$_2$O effectively hampers the excited state proton transfer to the imidazole N-atom. So, LIFM-CL1-H$_2$O displays blue emission at 463 nm in the solid state (Fig. 4) with higher quantum yield ($\Phi_{PL}$ = 22%, Supplementary Table 2) in the nature of the enol emission (*E**) of H$_2$hpi2cf. In contrast, the dehydrated LIFM-CL1 has no interference on intramolecular proton transfer, thus characteristic of keto emission (*K**) owing to normal ESIPT process ($E \to E^* \to K^* \to K \to E$), showing cyan emission at 493 nm with lower $\Phi_{PL}$ = 15%.

**Facile structure and PL switch for supersensitive water sensing.** The above reversible structural transformation and switchable dual emission endows the present Zn-MOF with capability as perfect model of water PL sensor. The fast hydration–dehydration structural interconversion can be easily achieved by vacuum or, even briefly, by gas blowing as observed from their emission change (Fig. 4). The blue emission of pristine LIFM-CL1-H$_2$O crystallites at 463 nm distinctly changes to cyan colour at 493 nm on pumping for ∼2 min, showing a red shift of ∼30 nm and a noticeable lowering of PL intensity and efficiency. Exposure of the evacuated sample to ambient air restores the original blue emission. The gases blowing test is applied by alternating different gas flows *in situ* onto the sample: pure dry gases of N$_2$, O$_2$, CO$_2$ and then wet gases of N$_2$ + H$_2$O, O$_2$ + H$_2$O, CO$_2$ + H$_2$O, as well as ambient air (RH = 45% at 25 °C). It is found that the blue emission of LIFM-CL1-H$_2$O tunes into cyan colour quickly

in a few seconds by blowing dry gases, and recovers the primary blue emission instantly on exposure to wet gases or air (Fig. 4 and Supplementary Fig. 4), indicating facile structural interconversion between the hydrated phase and the dehydrated phase. Such structural transformation is verified by *in situ* powder X-ray diffraction (PXRD) monitoring in air, vacuum and pure N$_2$ (Supplementary Fig. 5) representatively. The cycling tests under air-vacuum or air-N$_2$ conditions show well repeatable intensity switches between *E** and *K** emissions correlating to reversible SC–SC structural transformation (Fig. 4d) with slight luminescence loss after 10 cycles, demonstrating fast response, fatigueless reversibility and high photophysical stability of this PL-sensing material.

Another way to induce the water-driven structural and PL switch is the temperature variation. A gentle heating of LIFM-CL1-H$_2$O crystallites at 70 °C for a few seconds is enough to trigger PL change. And variable temperature PXRD patterns of the pristine LIFM-CL1-H$_2$O distinctly show the structural transformation process from hydrated phase to dehydrated LIFM-CL1 (Supplementary Fig. 6). In ambient atmosphere, the sample maintains the water-coordinated structure until 50 °C, where another set of diffraction peaks assignable to dehydrated phase appears. The total crystal transformation is completed at 75 °C, above which the structure remains unchanged and stable. On the contrary, the original hydrated phase is restored when cooling the sample to room temperature in air. Coexistence of hydrated and dehydrated phases during heating evidently indicates that the water escape/uptake can take place individually in a local part of the crystal without altering the rest crystal phase, thus denoting a nondestructive process to account for almost non-invasive water sensing by the Zn-MOF. Thermogravimetric (TG) curve of LIFM-CL1-H$_2$O also proves complete water loss (2.6 wt%) in between 50 and 112 °C, in good agreement with theoretical value (2.7 wt%) from the single-crystal analyses. Removal of coordinating DMF molecules is accompanied with the framework decomposition as confirmed by TG-mass spectrometry (MS) measurement (Supplementary Fig. 7).

It is a surprise that water escape/uptake proceeds in such a facile way taking the normal Zn–O$_{water}$ bond (2.027 Å) into consideration. Differential scanning calorimeter measurement (Supplementary Fig. 7c) reveals a rather low enthalpy required for complete water release from LIFM-CL1-H$_2$O (67.8 J g$^{-1}$ or 46.4 kJ mol$^{-1}$), only comparable to a mild H-bonding energy[53]. This may be owing to delicate arrangement of key atoms inside the micropores in LIFM-CL1-H$_2$O. As seen from Fig. 2 and Supplementary Fig. 3b, the water molecule resides closely to uncoordinating carboxylate O2-atom and hydroxyl O1-atom. During water-driven structural conversion, breaking of Zn–O$_{water}$

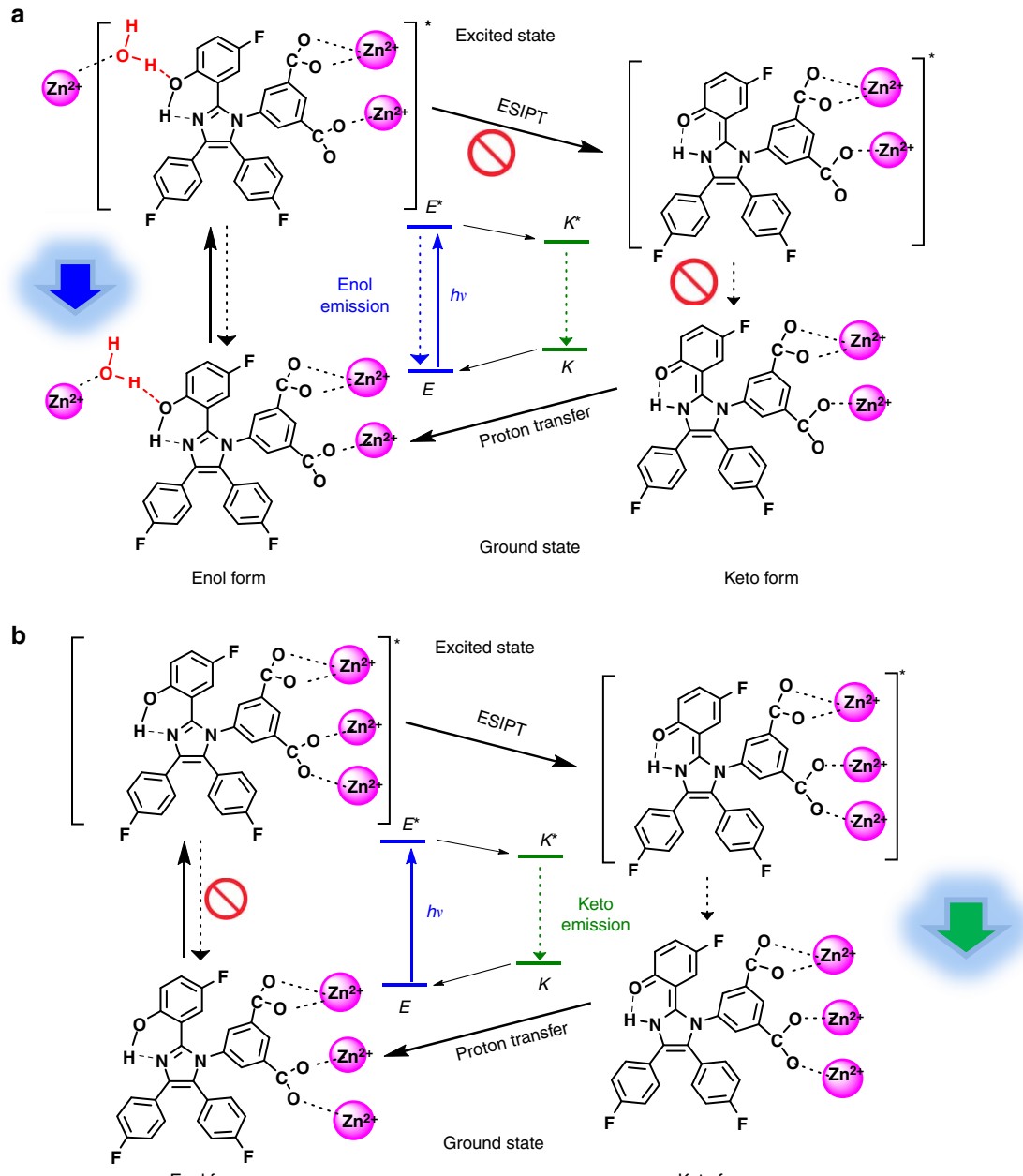

**Figure 3 | Luminescence switching mechanism.** Schematic representation of the water PL-sensing mechanism based on the different ESIPT processes in hydrated LIFM-CL1-H$_2$O (**a**) and dehydrated LIFM-CL1 (**b**) structures, in which the tautomerization between *E*-form and *K*-form on excitation is blocked in LIFM-CL1-H$_2$O by H-bonding, but turned on in LIFM-CL1.

bond is compensated locally by Zn–O2 bond formation, and *vice versa*, without significantly moving other atoms and influencing crystal packing. Meanwhile, the environment of –OH group is largely tuned with regard to ESIPT process. F-atoms are also situated inside the channel, which might deliberately balance the hydrophilic/hydrophobic attribute of the micropores, thus facilitating hydration/dehydration at rather mild conditions[54]. As shown from Supplementary Fig. 8, the water sorption isotherm at 298 K exhibits a type-I character with micropore filling exactly matching the theoretical uptake amount to convert dehydrated LIFM-CL1 into hydrated LIFM-CL1-H$_2$O (2.7 wt%), while a N$_2$ sorption isotherm at 77 K indicates minor N$_2$ uptake. The *in situ* emission spectral monitoring of dehydrated LIFM-CL1 exposed to EtOH or MeOH vapours (Supplementary Fig. 8) does not display recovery of the blue

emission as observed for wet gases. These adsorption and emission results suggest the micropores are just adequate for water diffusion, precluding any other small molecules for PL interference. The water uptake process measurement shows nearly linear adsorption kinetics up to equilibrium for water vapour at 298 K (Supplementary Fig. 9), giving a diffusion coefficient of $1.09 \times 10^{-6}$ cm$^2$ s$^{-1}$. The above characters of the microporous Zn-MOF represent unique selectivity and ultrafast responding speed compared with other reported samples (Supplementary Table 3)[55], and can be applied as a powerful water sensor which will be shown below.

**Film fabrication for potential applications.** To utilize the facile structure and PL transformation character of above Zn-MOF

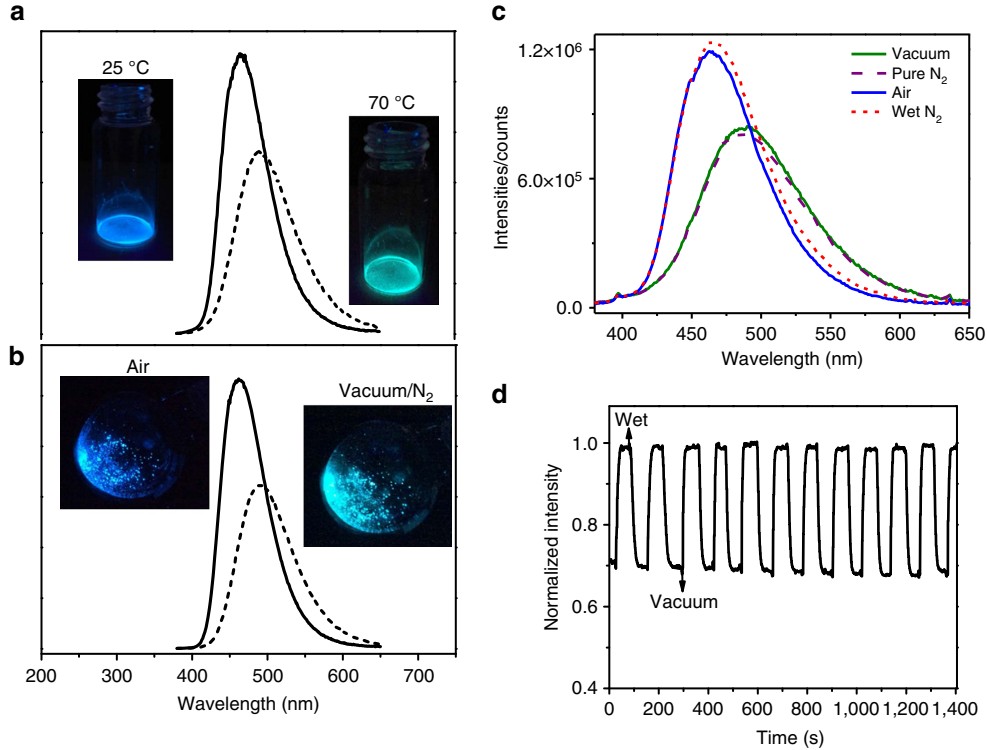

**Figure 4 | Photoluminescence switching.** (**a**–**c**) PL emission spectra and photographs of hydrated LIFM-CL1-$H_2O$ and dehydrated LIFM-CL1 tuned by heating, gases and vacuum, showing PL switch between blue *E*-emission and cyan *K*-emission based on ESIPT process of ligand $H_2$hpi2cf. (**d**) Time-dependent PL intensity cycles between LIFM-CL1 and LIFM-CL1-$H_2O$ microcrystals under vacuum and 1 bar air (RH = 45%) at 25 °C ($\lambda_{ex} = 365$ and $\lambda_{em} = 463$ nm).

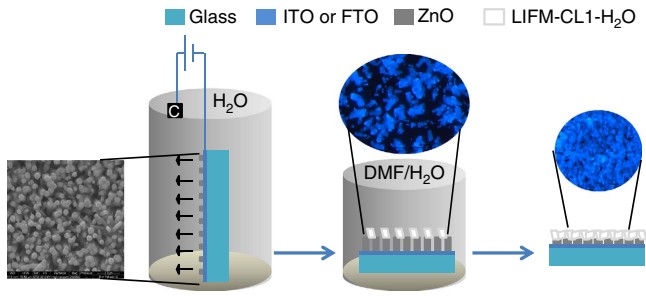

**Figure 5 | Film fabrication.** Schematic illustration of electrodeposition of and *in situ* growing LIFM-CL1-$H_2O$ onto ZnO-nanorod film, showing the SEM photo of the ZnO precursor and the fluorescence microscopy photos of the hybride LIFM-CL1-$H_2O$-ZnO films in different growing stages.

materials for water detection and thermo-inducing application in a more practical way, two types of LIFM-CL1-$H_2O$ films have been fabricated. One is paper-based film, which is prepared simply by coating the LIFM-CL1-$H_2O$ microcrystallines evenly on the filter paper to make a soft paper-supported film. And, another is ZnO-supported hybrid film obtained from *in situ* reaction of $H_2$hpi2cf with ZnO precursor on the surface of ZnO-nanorod film (Fig. 5)[56,57]. First, the supporting film of *Z* axis grown ZnO nanorods in several hundred micrometres is deposited on the indium tin oxide (ITO) or fluorine doped tin oxide (FTO) glass. Second, the precursor film is immersed in a clear DMF/$H_2O$ (1:1, v/v) solution of $H_2$hpi2cf at 100 °C, in which the ligands react with ZnO nanorods gradually. The solution is bubbled by air to modulate the formation of LIFM-CL1-$H_2O$ microcrystals instead of bulk crystals on the solid/ liquid interface. The growth of LIFM-CL1-$H_2O$-ZnO hybrid film

can be easily detected by appearance of blue fluorescence under 365 nm ultraviolet, which indicates that a dense LIFM-CL1-$H_2O$ film is formed in 10 min and verified further by PXRD tests (Supplementary Fig. 10). Finally, the *in situ* grown LIFM-CL1-$H_2O$-ZnO hybrid film is washed with DMF and water in turn, then dried in air.

Similar with the bulky crystals described above, both types of the LIFM-CL1-$H_2O$ films manifest fast and reversible PL switch between vacuum (cyan, 493 nm) and open air (blue, 463 nm) conditions (Supplementary Fig. 11), consistent with the structural transformation between dehydrated LIFM-CL1 and hydrated LIFM-CL1-$H_2O$, which has been confirmed by *in situ* PXRD tests in air and $N_2$ atmosphere (Supplementary Fig. 12). The cycling emission intensity tests at 463 nm under repeating air-vacuum or air-$N_2$ atmospheres also support the fully reversible PL interconversion without any fatigue effect (Supplementary Fig. 13).

To visually demonstrate the PL colour change accompanying the water release/uptake processes, both types of the LIFM-CL1-$H_2O$ films are blown by pure $N_2$ gas using a moving pipette and the videos are recorded under 365 nm ultraviolet light radiation (Supplementary Movies 1–3). As seen from the videos and Fig. 6, a cyan spot will appear at the $N_2$ outlet immediately in ∼2 s on the blue films placed in ambient conditions (RH = 45%, 25 °C), denoting fast local dehydration and PL switch. When moving the pipette to another site on the films, the original cyan spot is swiftly restored to blue while a new cyan spot turned on. These observations demonstrate how fast and facile the pinpoint structure and emission interconversions happen, and how easy and convenient the sensing films can release and absorb the water molecules into/from environmental atmosphere. Such kind of ultrafast and ultrasensitive response to atmosphere moisture is quite unique, and might finds practical applications in many

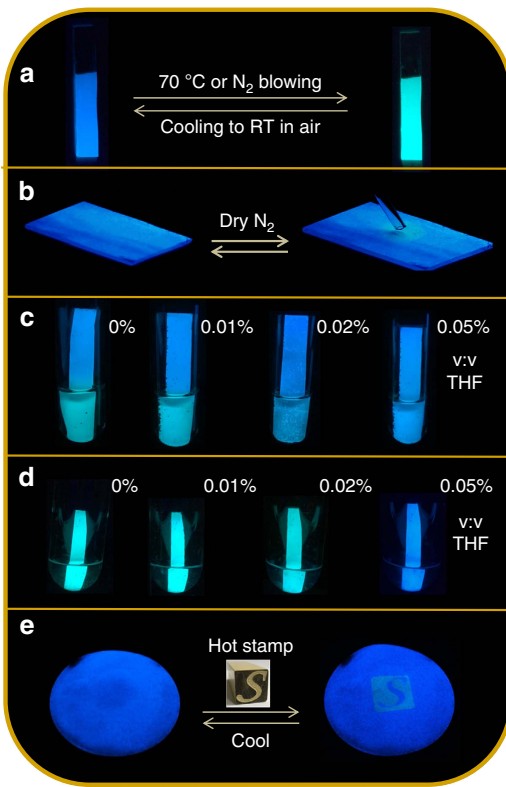

**Figure 6 | Practical application.** (**a**) Photographs showing PL colour switch of the ZnO-supporting hybrid film on heating or N$_2$ blowing and then cooling in air. (**b**) Photographs showing PL colour change of ZnO-supporting hybrid film under a moving pipette blowing pure N$_2$ (365 nm ultraviolet irradiation). (**c,d**) Photographs showing PL colour change by immersing hydrated (**c**) or dehydrated (**d**) ZnO-supporting hybrid films partly into THF solutions containing different amounts of water (0–0.05% v/v). (**e**) Photographs showing the thermal imaging using a stone seal (2 × 2 cm) with 'S' indenting symbol heated to 100 °C on the LIFM-CL1-H$_2$O paper film.

fields like on-site and real-time water sensing, or even imaging without any preactivation process of the sensing films required.

To estimate the capability of the Zn-MOF thin films for humidity sensing, the dehydrated LIFM-CL1 films are made by simply vacuumizing or N$_2$ purging of the as-prepared LIFM-CL1-H$_2$O films, and the emission spectra change in N$_2$ atmosphere with different RH of 0, 1, 12, 14, 17 and 21% are recorded (Supplementary Fig. 14). We can see that both the emitting wavelength and intensity are drastically changed in wet N$_2$ atmosphere, showing visual colour change from cyan to blue in ~2 s. A sensibility of moisture below RH 1% (lower limit of our hydrometer detection) is evident since the spectral and colour change can be observed even under this RH with the dehydrated films. Alternatively, the dry air atmosphere with RH at almost zero (pure N$_2$) can be detected by the hydrated films, which will show reverse colour change from blue to cyan. Therefore, the humidity sensing by the present Zn-MOF films is versatile because multiple detection methods, including PL intensity, wavelength and naked-eye observation, can be applied. Moreover, the detection can start from either hydrated or dehydrated films, which provide alternative choices.

For a practical detection of traces of water in organic solvents using present Zn-MOF as a PL sensor, the ZnO-supported hybrid films are applied in two ways: directly using hydrated LIFM-CL1-H$_2$O-ZnO film without preactivation, or using LIFM-CL1-ZnO film simply dehydrated by N$_2$ blowing or gentle heating at 70 °C

for a few minutes. As a demonstration shown in Fig. 6c,d and Supplementary Fig. 15, the ZnO-supported hybrid films are stuck into the testing solvents to rapidly judge their dryness by naked eye through observable emitting colour changing. The blue-emitting hydrated film turns to cyan immediately when touching the dry THF or CH$_3$CN solvents (0–0.01% v-v), corresponding to a dewatering process relative to structural transformation from LIFM-CL1-H$_2$O to LIFM-CL1 (representative emission spectra shown in Supplementary Fig. 16). In contrast, the organic solvents containing water above 0.05% v/v do not cause any colour change of the immersed hydrated film. Alternatively, the dehydrated film retains cyan emission when stuck into dry solvents, but turns blue in organic solvents containing water above 0.05% v/v, attributing to water uptake from solvents.

Furthermore, since water-driven structure and PL transformation can also be induced by temperature, the Zn-MOF thin films may find potential application in thermal imaging. As shown in Fig. 6a, the LIFM-CL1-H$_2$O-ZnO film exhibits distinct colour conversion between blue and cyan on heating and cooling in air. Therefore, a practical fast-responding demonstrator for thermal PL imaging is exemplified by a stone seal (2 × 2 cm) with 'S' indenting, which is heated to about 100 °C beforehand and then applied on the paper-based Zn-MOF film as thermal stimulus (~3 s) (Fig. 6e). The contact area turns cyan in accordance with the water-releasing transition, while gently shaking of the paper for cooling can easily erase the cyan 'S' as the emission recovers blue caused by the reverse water uptake from air. Such facile thermal imaging makes temperature-stimulated writing and tracing possible, which is convenient for confidential or green reusable purposes.

**Usage in thermometer and water contents determination.** As discussed above, the water molecules in the crystals of LIFM-CL1-H$_2$O can actually escape individually to lead to regional structure and PL change without influence on the whole crystal, and adsorption/desorption of water vapour follow an almost linear trendline until an equilibrium (Supplementary Fig. 9). This means the overall colour change of the similar sized crystals may be able to reflect the amount of the water release/uptake, thereof enabling establishment of certain relationship between emission variation and water molecules, which is useful for quantitative determination. As shown in Fig. 7a, when a batch of hydrated LIFM-CL1-H$_2$O microcrystals are heated in a stepwise way from 295 to 375 K, the emission spectra gradually turn from blue end (463 nm) to cyan end (493 nm), giving nearly a straight line of the colour change on the CIE coordinate diagram. Such linear correlation between CIE coordinates and temperature endows the present Zn-MOF crystals with potential as thermometer materials in this temperature range.

On the other hand, the content of traces of water in organic solvent may also be quantitatively determined. As shown in Fig. 7b and Supplementary Fig. 17, the emission spectra of a stirred suspension of dehydrated LIFM-CL1 microcrystals with different water concentrations in common organic solvents (EtOH, MeOH, DMF, THF, CH$_3$CN and acetone) are recorded. In general, all solvents display an overall enhancement of PL intensity with a concomitant blue shift on addition of aliquots of water. The measurements are performed instantly after adding the water aliquots into the organic solvents for intention of real-time water detection. In particularly, the water titration in EtOH and MeOH shows more continuous changes in the emission intensity compared with those in other organic solvents, but the emission maxima are shifted smaller, stopping at 476 and

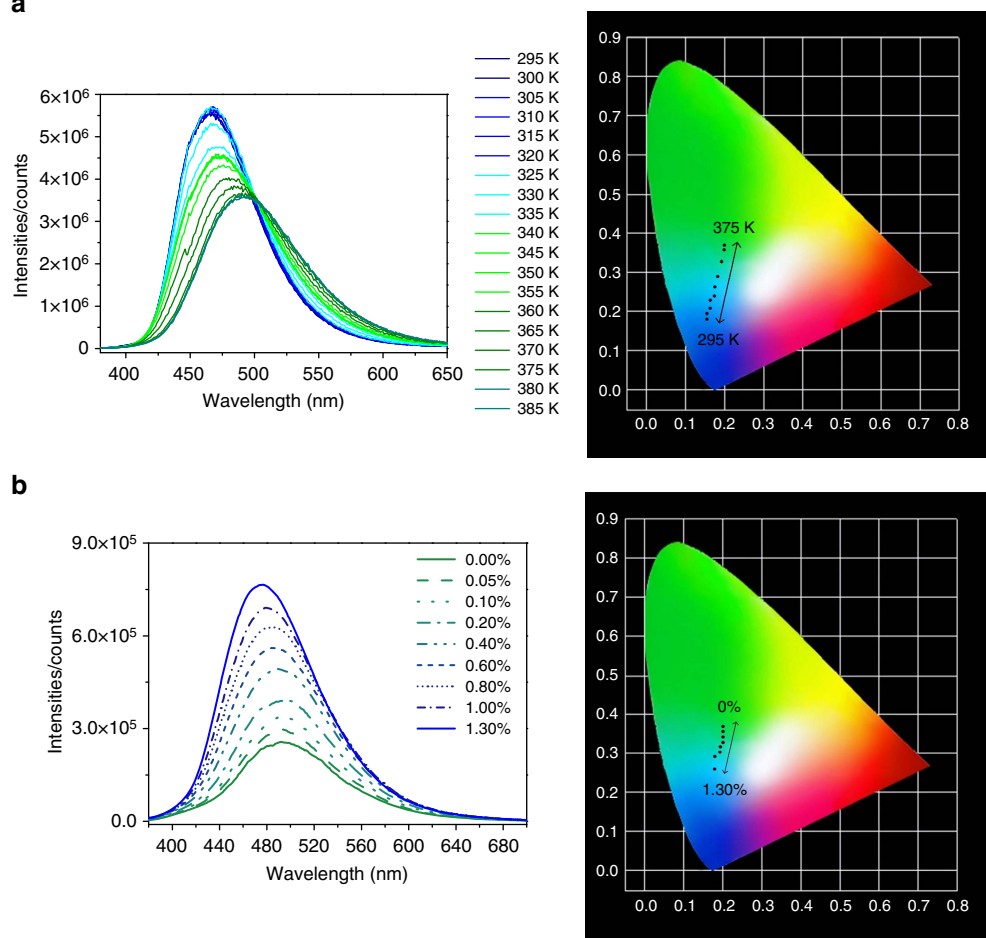

**Figure 7 | Application in thermometer and water content determination.** (**a**) Variable temperature emission spectra of hydrated LIFM-CL1-H$_2$O microcrystals in air showing a gradual transformation from hydrated LIFM-CL1-H$_2$O to dehydrated LIFM-CL1, and CIE coordinates corresponding to emission colour at each temperature. (**b**) Emission spectral change of a stirred suspension of dehydrated LIFM-CL1 microcrystals in dry MeOH solvents on addition of aliquots of water (% v-v) and corresponding CIE coordinates of emission colour. The excitation wavelength is 365 nm.

477 nm after the addition of 0.7% v-v and 1.3% v-v of water, respectively. Basically, a linear relationship of CIE coordinates versus water content, or, emission wavelength versus water content (Supplementary Fig. 18), can be established, which serves as a quantitative PL sensor for traces of water in EtOH and MeOH. In comparison, the PL behaviours are quite different in other solvents like acetone, CH$_3$CN and THF. The emission centre will restore 463 nm of the completely hydrated LIFM-CL1 when water concentrations reach to 0.25% v-v, 0.20% v-v and 0.40% v-v, respectively. The different PL responsive behaviours might be associated with different diffusion rate and uptake kinetics of water molecules in protic (MeOH and EtOH) and aprotic solvents (acetone, CH$_3$CN and THF). Correlation of PL emission with water content should be treated carefully depending on solvent nature and concentration range.

## Discussion

It is noteworthy that the structural interconversion in the present Zn-MOF is only driven by water removal/uptake and the emission switching is solely adjusted by water H-bonding. Therefore, LIFM-CL1 is consequently unique to water responsiveness avoiding any other solvents or gases interference. The fact that water removal/uptake causes only locally structural change (as shown in Fig. 6b and Supplementary Videos) without alteration of the whole crystal packing actually defines the water sensing in the Zn-MOF at

a molecular level. Namely, PL switching can occur in any detecting site on the sensor since each water molecule can separately drive pinpoint structural and emission change with nondestructive sensor recovery and lasting anti-fatigue ability (*vide infra*). Furthermore, compared with other reported luminescent humidity-sensing processes, especially those in MOFs and other porous materials, the water molecules are collected into the pore structure through adsorption, and basically, only when enough water molecules' adsorption are reached, can the luminescence change be tuned on, which determines the humidity measuring might need minutes to hours or even longer (Supplementary Table 3)[31–33,45,58,59]. While in our case, the molecular level-based detecting nature in the Zn-MOF enables ultrafast water detection in less than 2 s, which constitutes the fastest responding speed yet reported.

In summary, we have reported an ultrafast (~s level) and highly selective (1% RH or 0.05% v-v) water PL sensor based on a microporous Zn-MOF material featuring reversible water-driven SC–SC structural transformation and PL switch inherent in ESIPT process on a molecular level. Potential applications, including detection of humidity in gases, traces of water in organic solvents, thermal imaging and thermometer, have been applied by preparation of microcrystals and paper-based or ZnO-supporting hybrid films. The water-sensing behaviours of the present Zn-MOF material are versatile, that is, starting from either hydrated or dehydrated materials, detected by either emission wavelength, intensity or CIE coordinate, stimulated via

either temperature or water contents in gases or liquids. On the other hand, the water sensing is unique, because the microporous Zn-MOF is responsive solely to water excluding interference from any other small molecules. Furthermore, the water release/uptake can occur easily, in an ultrafast way and individually to cause only pinpoint structural change and local PL switch without influencing the whole sensor phase, thus making the sensing and imaging processes almost nondestructive without fatigue for activation and recovery, and showing the fastest responding speed, thereof suitable for instant tracing, writing, patterning, displaying and so on. The current work provides not only a very rare example of supersensitively accessible SC–SC transformation in MOF materials, but also a facile and widely applicable approach for development of a new class of chemo- and thermo-sensors.

## Methods

**General considerations.** All reaction materials were obtained from commercial suppliers, and used without further purification.

$^1$H NMR spectra was obtained by Bruker AVANCE III (400 MHz) in DMSO-$d_6$ solutions. Elemental analysis was carried out using a vario EL cube elemental analyzer. Fluorescence spectra were measured on an Edinburgh FLS980 or FLS5 Photoluminescence Spectrometer. Absolute PL quantum yields were measured with emission scans that were further processed using the quantum yield wizard provided by the Quantaurus-QY (Hamamatsu, Japan) in the integrating sphere, and the quantum yield of LIFM-CL1-$H_2O$ was tested from the heated LIFM-CL1 powder (120 °C) in air. Estimated experimental error for quantum yields determination is 1%. The fluorescence lifetime experiments were performed in the time-correlated single photo counting methods by using a 350 nm picoseconds pulsed diaed laser with a repetition rate of 10 MHz (100 ns) as the excitation source. PXRD measurements were performed on an X-ray powder diffractometer (Rigaku, Japan), operating at 4 kW, 3 mA. Crystal data collection was performed on a Super Nova X-ray diffractometer system (Agilent Technologies, America). The RH was determined by an AM2303 digital humidity and temperature sensor. The morphologies and structures of MOFs were characterized by SU8010 field emission scanning electron microscopy (SEM) (Hitachi, Japan, Supplementary Fig. 19). The further confirmation of water content of organic solvents was conducted on C30 Karl Fischer Coulometric Titrimetry (METTLER TOLEDO, Switzerland, Supplementary Tables 4 and 5). The stability of the Zn-MOF samples in different organic solvents was also confirmed by PXRD tests (Supplementary Fig. 20).

Differential scanning calorimeter measurement was performed on a TA Q2000 instrument at a heating/cooling rate of 10 K min$^{-1}$. The thermostability was studied by TG-MS devices (NETZSCH, Germany). Water vapour sorption and kinetic isotherms were measured with an intelligent gravimetric sorption analyzer (Hiden-IGA100B, UK) at 298 K. Before the sorption experiments, the as-synthesized sample was placed in the sample tube and dried under high vacuum at 50 °C for 6 h. The $N_2$ sorption properties were measured with a gas sorption surface area and pore size analyser (QUADRASORB evo, USA) at 77 K. Before the sorption experiments, the as-synthesized sample was placed in the sample tube and dried under high vacuum at 50 °C for 6 h. The calculation of chemical diffusion coefficient was performed using the IGA software for sorption-time data recorded during isotherm measurement; equation for calculation of diffusion coefficient: $M_t/M_\infty = 1 - \sum_{n=1}^{\infty} \frac{6}{[n\pi]^2} \exp\left[\frac{-D_{chem}n^2\pi^2(t-t_0)}{h^2}\right]$, $D_{chem}$ is diffusion coefficient, $M_t$ is the amount of diffusing species at time $t-t_0$ and the sample is in equilibrium at time $= t_0$ and then relaxes to equilibrium $M_\infty$ under constant chemical potential, $h$ is the sphere radius of the sample.

**Synthesis of H$_2$hpi2cf ligand.** The reagent was acquired as previously reported by our group[39] (see Supplementary Fig. 1 for the synthetic route and Supplementary Methods for details).

**Synthesis of LIFM-CL1-H$_2$O [Zn(hpi2cf)(DMF)(H$_2$O)] crystals.**
Zn(NO$_3$)$_2$·6H$_2$O (0.03 g, 0.1 mmol), H$_2$hpi2cf (0.053 g, 0.1 mmol) and 4 ml DMF/H$_2$O (v/v = 1:1) were added into a 10 ml Teflon cup. The mixture was stirred for 5 min and then the Teflon cup was incased into the matched stainless steel autoclave. The sealed autoclave was heated at 100 °C in the oven for 50 h and cooled to the room temperature at the rate of 10 °C h$^{-1}$. Colourless, block crystals of LIFM-CL1-H$_2$O were gained by filtration and dried in vacuum. Yield: 0.04 g. Anal. calcd. for C$_{32}$H$_{24}$F$_3$N$_3$O$_7$Zn (%): C, 56.11; H, 3.53; N, 6.14; found: C, 56.04; H, 3.346; N, 6.01.

**Synthesis of LIFM-CL1-H$_2$O microcrystallines for film preparation.**
Zn(NO$_3$)$_2$·6H$_2$O (0.1 g, 0.3 mmol), H$_2$hpi2cf (0.106 g, 0.2 mmol), and 8 ml DMF/H$_2$O (1:1 v/v) were added into a 15 ml glass tube with pressurized and thick

wall, which was sealed and heated at 110 °C for 5 h and the mixture was stirred in high speed at the same time. After cooling to room temperature, the liquid was separated by centrifuging. The resultant fine white powder was washed by DMF for three times to remove the unreacted ligand, and then washed by ethyl acetate for several times, and finally dried in vacuum at room temperature for 4 h. Yield: 0.135 g. The powder was subjected to PXRD measurement for determination of LIFM-CL1-H$_2$O character. The morphology was examined using SEM, which reveals generally alike crystals with size in several micrometres.

**Synthesis of ZnO-nanorod precursor film.** The synthesis was carried out by electrodeposition as reported[56]. The FTO or ITO glass was cleaned in ultrasonic bath by deionized water, then ethanol and finally rinsed by deionized water again before electrodeposition. The ZnO-nanorod arrays were prepared via cathodic electrodeposition in a solution containing 0.02 M Zn(NO$_3$)$_2$ + 0.01 M CH$_3$COONH$_4$ + 0.01 M (CH$_2$)$_6$N$_4$ with a current density of 0.5 mA cm$^{-2}$ at 90 °C for 1 h. The as-synthesized ZnO-nanorod arrays were dried at room temperature in atmosphere.

**Sensing experiments.** As small, confined space helps to stabilize the humidity more quickly, the test was measured in the OptistatDN2 thermostat (OXFORD-instruments, UK), which was originally used for test of temperature-dependence PL spectrum. The humidity probe, a small electronic device, was placed in the sample cavity, connected by a tenuous wire, which was hooked to the LED screen outside that displayed the real-time RH value. Nitrogen with different moisture content gently blew into the sample cavity through a fine duct and some gaps were left for gas exchange to reach a certain RH value. When blowing pure N$_2$, the RH would drop to 1% in about 1 min. No more decrease could be detected due to the lowest limit of the device. To gain different humidity, the pure N$_2$ was purged through H$_2$SO$_4$ containing different content of water. All measurements were carried at environment temperature (20 °C).

**Detection of traces of water in organic solvents.** As the Zn-MOF existed as LIFM-CL1-H$_2$O form in the solid state in air at room temperature, so before the sensing test, the blue emitting (under 365 ultraviolet) LIFM-CL1-H$_2$O (2 mg) was immersed in the dehydrated organic solvent and the powder in the liquid immediately changed to cyan-emitting colour, demonstrating the complete SC–SC transformation to the dehydrated LIFM-CL1 form. The clear supernatant was carefully removed and new dry organic solvent was added to get 3 ml suspended solution, and then, traces of water was added using a precision micropipette (0.1–10 μl range) to achieve the desired water concentration. The whole process was operated in a 3.5 ml luminescence cuvette. After each addition, the cuvette should be sealed to isolate the water in surrounding air and emission spectra were recorded instantly. The system was kept in suspension by continuous stirring using the magnetic stirring accessory of the instrument. The emission spectrum after each addition was recorded three times to ensure signal stability.

**Data availability.** The data that support the findings of this study are available from the corresponding author on request. The X-ray crystallographic coordinates for structures reported in this study have been deposited at the Cambridge Crystallographic Data Centre (CCDC), under deposition numbers 1520199–1520202. These data can be obtained free of charge from The Cambridge Crystallographic Data Centre via www.ccdc.cam.ac.uk/data_request/cif.

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

## Acknowledgements

This work was supported by NSFC (21373276, 91222201 and 21573291), STPP of Guangzhou (201510010246 and 201504010031) and NSF of Guangdong Province (S2013030013474). We thank Mr Yuan-Jian Ye from Guangzhou Quality Supervision and Testing Institute for the determination of water content in organic solvents; and Dr Rui-Ping Chen from Fujian Institute of Research on the Structure of Matter for the water vapour adsorption and kinetic measurement.

## Author contributions

C.-Y.S. and M.P. designed the research and wrote the paper. L.C. carried out most of the syntheses and measurements. J.-W.Y., H.-P.W., S.-Y.Y., Z.-W.W., L.-Y.Z., K.W. and Y.-N.F. helped in some experiments and data analyses. All authors discussed the results and commented on the manuscript.

**Additional information**

**Competing interests:** The authors declare no competing financial interests.

