## [Peer Review File · Nature Communications]

Reviewers' comments:

Reviewer #1 (Remarks to the Author):

The paper describes a Zn²⁺ coordination polymer which shows an interesting reversible hydration-dehydration property and importantly, this can be done in a single-crystal-to-single-crystal fashion by just blowing N₂ or under vacuum. Furthermore, this property can be used for water detection in organic solvents or in gas phase because of the different luminescence of the hydrated vs. the dehydrated form. This is an extremely sensitive water sensor with very fast response. This paper is of significant novelty and is suitable for publication in Nature Communications. I have only one suggestion: Please write the exact formulas of hydrated and dehydrated compounds in the introduction.

Reviewer #2 (Remarks to the Author):

This study focuses on luminescence based sensing of water by LIFM-CL1, a Zn-based MOF, specifically designed and assembled by employing a dual-emissive ligand (H₂hpi2cf) that exhibits excited state intramolecular proton transfer (ESIPT). The dual-emissive behavior of the resulting MOF allows effective and selective detection of water by changes in color, rather than emission intensity. Detection experiments were performed under various conditions, including dry/wet gases and organic solvents, in air and under nitrogen. The facile and reversible structural transformation between dehydrated and hydrate form gives rise to ultrafast and sensitive detection of water. The structural transformation through water desorption/adsorption is also sensitive to heat, so the MOF may also be potentially useful in thermal imaging. A detailed analysis was provided on sensing mechanism. In addition, films in two forms were fabricated and their sensing performance was evaluated. The work introduces a new, simple, and efficient method for trace water detection and may be promising for a variety of applications in the industrial, medical and environmental fields. The following points should be fully addressed before the paper may become suitable for publication in Nat. Comm.

1. The dual emission mechanism of the MOF was attributed to ligand-based ESIPT process that involves the transfer of the hydroxyl proton to the electronegative atom via an intramolecular hydrogen bond. To prove this, the authors need to provide and discuss in the revised manuscript the O-H...N and O...H-N distances in the LIFM-CL1-H₂O and LIFM-CL1 structures, respectively, to demonstrate that changes in these atomic distances fully correlate with the E- and K-form of the ligand.
2. What actions have been made to confirm that the shift in emission energy (463 < 493 nm) is solely due to the inhibition/activation of the ESIPT pathway? Even a small structural alternation, such as changing the local coordination of Zn atoms and transforming from a series of 1D chains to 2D layers, could induce emission changes due to relative positions of emissive ligand, preferred orientations for absorption of light, etc. which would be independent of the proton transfer pathways.
3. Additional data may be necessary for the sensing analysis of the MOF in organic media (e.g. THF, MeOH, CH₃CN). Is it possible that the organic solvents exchange with DMF? Since DMF evacuation prompts a framework breakdown, solvent exchange could trigger the same outcome, PXRD patterns should be taken after the sensing experiments, or collected in-situ throughout the exposure (similar to those in Supplementary Figure 6).
4. What is the detection limit (in RH) of the MOF? A comparison of its detection limit with benchmark water sensory materials would be very useful.
5. Along the same line, linear relationship between emission wavelength and water content was

discussed on pg. 14 (Lines 273-275). These data can be used to obtain a quantitative performance measure of the MOF sensor for detecting traces of water in organic solvents by carrying out a Stern-Volmer analysis using the information in Supplementary Figure 18.

6. Does the LIFM-CL1 water sensor also respond to other small polar molecules that are capable of coordinating to Zn and forming H-bonds? This is important to claim its water selectivity.

7. Supplemental Figure 5 should also include PXRD patterns of sample exposed in wet gas mixtures.

8. Supplemental Figure 8: Further explanation is needed on water desorption. Typically Zn-MOFs lose coordinated water at elevated temperatures (e.g. 100 - 200 C). This is indeed the case in this study (see Supplementary Figure 7). While partial removal of coordinated water molecules is possible by solely evacuation at room temperature, it is unlikely that full removal of water can be achieved under such conditions. How long was the desorption process? This information should be included.

9. Pg. 9, lines 168-169 and Supplemental Figure 9: A brief description on the kinetic model and calculations of diffusion coefficient should be provided.

Minor points:

1. Typos: (a) pg. 8, line 152, "the the" should be "the". (b) pg. 12, line 232, "LIFM-CL1 to LIFM-CL1-H2O" should be "LIFM-CL1-H2O to LIFM-CL1". (c) Figure captions: "upper", "lower" should be "top" and "bottom". (d) Supplemental Figure 12: The PXRD pattern after N2 blowing shows a mixture of both forms. Shouldn't it match with the dehydrated form? Likewise, the PXRD pattern after exposure to air should match the LIFM-CL1-H2O, not LIFM-CL1.

2. Supplemental Figure 4: The notation (a) and (b) are not given in the Figure caption.

Reviewer #3 (Remarks to the Author):

In this manuscript, Su and coworkers described the design and preparation of a microporous Zn-MOF (LIFM-CL1) and corresponding films for water sensing and thermal imaging. The single-crystal-to-single-crystal (SC-SC) transformation driven by reversible removal/uptake of coordinating-water molecules endows this sensor ultrafast response and high selectivity. The experiments are well designed and the diagrams are adequately used to report the relevant results in a clear way. In my opinion, this paper would be accepted for nature communications once the following questions have been addressed clearly:

1. The equation of SC-SC transformation ($\text{LIFM-CL1-H}_2\text{O} \leftrightarrow \text{LIFM-CL1}$) is improper because the removal water is missed on the right; " $\text{Zn}(\text{NO}_3)_2$ " in Figure 1 should be " $\text{Zn}(\text{NO}_3)_2 \cdot 6\text{H}_2\text{O}$ "; Page 12 Line 231, "structural transformation from LIFM-CL1 to LIFM-CL1-H2O" might be "structural transformation from LIFM-CL1-H2O to LIFM-CL1"; Page 2 Line 46 in Supplementary Figure 6, "Zn-PCP" should be "Zn-MOF".

2. Page 7 Line 129, the authors say "reversible SC-SC structural transformation (Fig. 4d) without any luminescence loss after 10 cycles", however, as shown in Fig. 4d, the luminescence intensities in vacuum gradually decrease, so the description is not quite exact. Moreover, as shown in Fig. 4d and Supplementary Figure 13, the powder and film of LIFM-CL1-H2O show different quenching efficiency and different recovery time under the same conditions, why?

3. The authors present the Zn-MOF can remove water from solvents or dewater to solvents. In order to more clearly illustrate this point, the water content of solvents before and after being

soaked with hydrated LIFM-CL1-H₂O and dehydrated LIFM-CL1 should be confirmed with other methods such as Karl Fischer titration or electrochemical methods. The water contents in fluorescence titration experiments also should be confirmed with other methods.

4. The single-crystal-to-single-crystal (SC-SC) transformation is carried out in accordance with the stoichiometric ratio, such as each Zn²⁺ needs one water, so I think the concentration of sensor might influence the detective results and the excess of sensor or water content might give different results. The author should conduct some experiments about it. Moreover, we don't know the water content of real samples, so how to choose the concentration of sensor?

To Reviewer #1

Comments:

The paper describes a Zn^{2+} coordination polymer which shows an interesting reversible hydration-dehydration property and importantly, this can be done in a single-crystal-to-single-crystal fashion by just blowing N_2 or under vacuum. Furthermore, this property can be used for water detection in organic solvents or in gas phase because of the different luminescence of the hydrated vs. the dehydrated form. This is an extremely sensitive water sensor with very fast response. This paper is of significant novelty and is suitable for publication in Nature Communications. I have only one suggestion: Please write the exact formulas of hydrated and dehydrated compounds in the introduction.

Response: Thank for reviewer's quite positive appreciation of this work. As suggested, we have added the exact formulas of hydrated and dehydrated compounds in the introduction. The hydrated one: $\text{Zn}(\text{hpi}2\text{cf})(\text{DMF})(\text{H}_2\text{O})$; the dehydrated one: $\text{Zn}(\text{hpi}2\text{cf})(\text{DMF})$.

To Reviewer #2

Comments:

This study focuses on luminescence based sensing of water by LIFM-CL1, a Zn-based MOF, specifically designed and assembled by employing a dual-emissive ligand ($\text{H}_2\text{hpi}2\text{cf}$) that exhibits excited state intramolecular proton transfer (ESIPT). The dual-emissive behavior of the resulting MOF allows effective and selective detection of water by changes in color, rather than emission intensity. Detection experiments were performed under various conditions, including dry/wet gases and organic solvents, in air and under nitrogen. The facile and reversible structural transformation between dehydrated and hydrated form gives rise to ultrafast and sensitive detection of water. The structural transformation through water desorption/adsorption is also sensitive to heat, so the MOF may also be potentially useful in thermal imaging. A detailed analysis was provided on sensing mechanism. In addition, films in two forms were fabricated and their sensing performance was evaluated. The work introduces a new, simple, and efficient method for trace water detection and may be promising for a variety of applications in the industrial, medical and environmental fields. The following points should be fully addressed before the paper may become suitable for publication in Nat. Comm.

1. The dual emission mechanism of the MOF was attributed to ligand-based ESIPT process that involves the transfer of the hydroxyl proton to the electronegative atom via an intramolecular hydrogen bond. To prove this, the authors need to provide and discuss in the revised manuscript the O-H...N and O...H-N distances in the LIFM-CL1-H₂O and LIFM-CL1 structures, respectively, to demonstrate that changes in these atomic distances fully correlate with the E- and K-form of the ligand.

Response: We are sorry to make the reviewer confused probably due to our unclear discussion. As a matter of fact, the ground state of both LIFM-CL1-H₂O and LIFM-CL1 are in enol form, as confirmed by the single-crystal structural analyses, in which the electron cloud of H atom is much closer to O than to N in both structures. The **intramolecular** hydrogen bonds are formed in both cases, of which the O...N distance is 2.552 Å in the hydrated crystal and 2.606 Å in the dehydrated crystal. However, in the hydrated structure, the **intermolecular** hydrogen bond is also formed due to the existence of the adjacent coordinating-water molecule (O6). Such intermolecular hydrogen bond is absent in the dehydrated structure. *This difference in intermolecular hydrogen bonds between the hydrated and dehydrated structures leads to distinct ESIPT processes upon structural transformation as we discussed in the main text.* Now we add the sentence “**From the structural determination, the hpi2cf²⁻ ligands in both LIFM-CL1-H₂O and LIFM-CL1 are in enol form. However, the intermolecular H-bonding difference imparts distinct effect on the ESIPT processes in these two interconvertible structures**” in the text to make the above statement clearer.

To approve the existence of only enol form in the ground state of ESIPT molecules, the following publications can be referenced, such as *Adv. Mater.*, 2011, **23**, 3615–3642; *Phys. Chem. Chem. Phys.*, 2012, **14**, 8803–8817. As stated in these references, the widely recognized ESIPT process is as following: the ESIPT molecules exist exclusively in enol (E) form in the electronic ground state. While photoexcitation causes the tautomeric transformation from the excited enol form (E*) to the excited keto form (K*) in sub-picosecond time scale. After decaying radiatively to the ground state, reverse proton transfer recovers the initial E form (Figure 3 and Supplementary Figure 3). So K form is in only available in the higher level excited state as a metastable intermediate while not in the stable ground state.

2. What actions have been made to confirm that the shift in emission energy (463 < 493 nm) is solely due to the inhibition/activation of the ESIPT pathway? Even a small structural alternation, such as changing the local coordination of Zn atoms and transforming from a series of 1D chains to 2D layers, could induce emission changes due to relative positions of emissive ligand, preferred orientations for absorption of light, etc. which would be independent of the proton transfer pathways.

Response: Thank for the kind reminding. We agree with that even a small structural alternation could lead to emission changes. However, the case in our work is particular. During the SC-SC transition from LIFM-CL1-H₂O to LIFM-CL1, only delicate structural changes happen individually in a local microenvironment to remove the coordinating water molecule and form the Zn-O2 bonding, while all other atoms are not influenced and the overall crystal packing is perfectly persisted. Therefore, the transformation from 1D chains to 2D layers does not change the relative position of the framework atoms, especially the aromatic emissive component of the ligand. Moreover, such structural change has little effect on the preferred orientations of the ligand for absorption of light, etc.

Besides, we have confirmed that the shift in emission energy (463 \leftrightarrow 493 nm) is due to the ESIPT process in two interconvertible structures (LIFM-CL1-H₂O \leftrightarrow LIFM-CL1) because we had studied the luminescent property of H₂hpi2cf molecule (organic ligand in this work) in detail. In the published result (Ref 39), B-form and C1-form of H₂hpi2cf showed the same molecular configuration but different emission energy (B, 463 nm; C1, 486 nm). Distinguishingly, in B-form crystals, the ESIPT active-site formed intermolecular H-bonds with the oxygen atom of the ethanol solvent molecule, which hampered the intramolecular ESIPT reaction within H₂hpi2cf, and ultimately caused predominant E* emission with higher energy. While in C1-form, no obvious intermolecular forces and hydrogen bonds were formed around the ESIPT active-site, so the emission was typically K*-related and showed a red shift compared with the B-form (See the following figure).

3. Additional data may be necessary for the sensing analysis of the MOF in organic media (e.g. THF, MeOH, CH₃CN). Is it possible that the organic solvents exchange with DMF? Since DMF evacuation prompts a framework breakdown, solvent exchange could trigger the same outcome, PXRD patterns should be taken after the sensing experiments, or collected in-situ throughout the exposure (similar to those in Supplementary Figure 6).

Response: Thank for the reviewer's suggestions. The structure of the present Zn-MOF is rather unique due to its tiny micropore nature and excellent stability. Therefore, organic solvents such as THF, MeOH, CH₃CN, or DMF, etc, cannot enter the crystal channels, and the framework breakdown and solvent exchange will not happen. In our original manuscript, we have confirmed that the dehydrated LIFM-CL1 even cannot adsorb N₂ gas, and shows no emission change upon test to EtOH or MeOH vapors.

As suggested by the reviewer, we further confirm this by adding the following tests: the LIFM-CL1-H₂O powder (ground sample from crystals) is immersed into organic solvents (CH₃CH₂OH, CH₃CN, CH₃OH, DMF, THF) for two hours, and then the powder is tested for PXRD. The result is shown in Supplementary Figure 20, which shows that the immersed MOF has no phase change, demonstrating the high stability of LIFM-CL1-H₂O in common organic solvents and the feasibility in the following water sensing analysis in organic solvents.

4. What is the detection limit (in RH) of the MOF? A comparison of its detection limit with benchmark water sensory materials would be very useful.

Response: From our experimental results, the detection limit of the Zn-MOF is less than RH 1% (20 °C). It is hard to exactly manage an even lower RH tests based on our experimental conditions. We have referred to other sensory materials as shown in Supplementary Table S3. It is clear that our report constitutes an excellent sensory material well outperforming other known water sensory materials.

5. Along the same line, linear relationship between emission wavelength and water content was discussed on pg. 14 (Lines 273-275). These data can be used to obtain a quantitative performance measure of the MOF sensor for detecting traces of water in organic solvents by carrying out a Stern-Volmer analysis using the information in Supplementary Figure 18.

Response: Thank for the good suggestion. Stern-Volmer analysis is usually used in sensing based on fluorescence quenching (Ref: *Chem. Soc. Rev.*, 2014, **43**, 3666-3761; *Anal. Chem.*, 1995, **67**, 1377-1380). In our work, the fluorescence switching is based on the interconvertibility between two structures, so the Stern-Volmer equation is not quite appropriate for this kind of analysis. Nevertheless, from some other reports about structural conversion related luminescent sensing (Ref: *Chem. Sci.*, 2011, **2**, 2214-2218; *Nat. Mater.*, 2011, **10**, 787-793; *Angew. Chem. Int. Ed.*, 2013, **52**, 710-713; *Chem. Commun.*, 2014, **50**, 1444-1446; *CrystEngComm* 2012, **14**, 7157-7160; *Nat. Mater.*, 2011, **10**, 787-793), we can see that the relationship between the concentration of detected component and emission identity are used to give a quantitative performance for the sensing. Therefore, the basically linear relationship between the concentration of detected water and emission position of our MOF material given in Supplementary 18 can indeed serve as a quantitative sensor for detecting traces of water in organic solvents. However, it should be noted that the accuracy and quantification of such kind of sensing is dependent on the precise control of the detecting conditions and exact equilibrium of water uptake by the sensing materials, and thereof should not be emphasized here without strict analytical conditions.

6. Does the LIFM-CL1 water sensor also respond to other small polar molecules that are capable of coordinating to Zn and forming H-bonds? This is important to claim its water selectivity.

Response: Thank for the kind reminding. As we respond above, the unique microporous nature of LIFM-CL1 excludes possibility of adsorbing other small polar molecules with size larger than water. We have tested and compared a series of small molecules. Just as Supplementary Figure 8 (the lower emission spectra) shows, the dehydrated LIFM-CL1 sample does not respond to methanol or ethanol molecules. And also, this MOF does not respond to other common organic solvents like CH₃CN, THF and DMF. In general, LIFM-CL1 has unique water selectivity in common organic solvent detecting conditions.

7. Supplemental Figure 5 should also include PXRD patterns of sample exposed in wet gas mixtures.

Response: As suggested, we add the PXRD test in the condition of 98% RH, air atmosphere (Supplementary Figure 5b), which shows the Zn-MOF persists in LIFM-CL1-H₂O phase.

8. Supplemental Figure 8: Further explanation is needed on water desorption. Typically Zn-MOFs lose coordinated water at elevated temperatures (e.g. 100 - 200°C). This is indeed the case in this study (see Supplementary Figure 7). While partial removal of coordinated water molecules is possible by solely evacuation at room temperature, it is unlikely that full removal of water can be achieved under such conditions. How long was the desorption process? This information should be included.

Response: Thank for the reviewer's advice. Indeed, typical MOF needs higher temperature to lose the coordinated water molecules, and a long time (about ten minutes in the measurement of Supplementary Figure 8) will be needed for a total water desorption process. While in our experiment, partial instead of complete removal of coordinated water molecules by solely evacuation is enough to trigger the corresponding luminescent sensing behavior, and this just needs a rather short time scale within seconds.

9. Pg. 9, lines 168-169 and Supplementary Figure 9: A brief description on the kinetic model and calculations of diffusion coefficient should be provided.

Response: We add a brief description for the kinetic model and calculation of diffusion coefficient in the methods part of the main text:

The calculation of chemical diffusion coefficient was performed using the IGA software for sorption-time data recorded during isotherm measurement; equation for calculation of

diffusion coefficient: $M_t/M_\infty = 1 - \sum_{n=1}^{\infty} \frac{6}{[n\pi]^2} \exp\left[\frac{-D_{chem}n^2\pi^2(t-t_0)}{h^2}\right]$, D_{chem} is diffusion coefficient, M_t is the amount of diffusing species at time $t-t_0$ and the sample is in equilibrium at time= t_0 and then relaxes to equilibrium M_∞ under constant chemical potential, h is the sphere radius of the sample.

Minor points:

1. Typos: (a) pg. 8, line 152, “the the” should be “the”. (b) pg. 12, line 232, “LIFM-CL1 to LIFM-CL1-H₂O” should be “LIFM-CL1-H₂O to LIFM-CL1”. (c) Figure captions: “upper”, “lower” should be “top” and “bottom”. (d) Supplemental Figure 12: The PXRD pattern after N₂ blowing shows a mixture of both forms. Shouldn’t it match with the dehydrated form? Likewise, the PXRD pattern after exposure to air should match the LIFM-CL1-H₂O, not LIFM-CL1.

Response: Many thanks for the kind suggestions. We have made the following alterations accordingly:

(a) (b) (c), **We have corrected them in the revised manuscript.**

(d), The equipment of the N₂ blowing PXRD measurement is shown as the following image. Because the nozzle we used for N₂ flow is very thin, so not the entire surface of LIFM-CL1-H₂O-ZnO hybrid film can be blown and led to structural transformation. Therefore, the part of the film not blown by N₂ flow will keep the LIFM-CL1-H₂O phase, while this part is also exposed to X-ray beam and the PXRD pattern is collected together. And finally the PXRD pattern after N₂ blowing shows a mixture of the two forms.

2. Supplemental Figure 4: The notation (a) and (b) are not given in the Figure caption.

Response: We add the notation in the Figure caption of Supplementary Figure 4.

To Reviewer #3

Comments:

In this manuscript, Su and coworkers described the design and preparation of a microporous Zn-MOF (LIFM-CL1) and corresponding films for water sensing and thermal imaging. The single-crystal-to-single-crystal (SC-SC) transformation driven by reversible removal/uptake of coordinating-water molecules endows this sensor ultrafast response and high selectivity. The experiments are well designed and the diagrams are adequately used to report the relevant results in a clear way. In my opinion, this paper would be accepted for nature communications once the following questions have been addressed clearly:

1. The equation of SC-SC transformation ($\text{LIFM-CL1-H}_2\text{O} \leftrightarrow \text{LIFM-CL1}$) is improper because the removal water is missed on the right; “ $\text{Zn}(\text{NO}_3)_2$ ” in Figure 1 should be “ $\text{Zn}(\text{NO}_3)_2 \cdot 6\text{H}_2\text{O}$ ”; Page 12 Line 231, “structural transformation from LIFM-CL1 to LIFM-CL1-H₂O” might be “structural transformation from LIFM-CL1-H₂O to LIFM-CL1”; Page 2 Line 46 in Supplementary Figure 6, “Zn-PCP” should be “Zn-MOF”.

Response: Thank for the kind suggestions. We have made the following corrections accordingly:

- (1) The equation of SC-SC transformation is revised as $\text{LIFM-CL1-H}_2\text{O} \leftrightarrow \text{LIFM-CL1} + \text{H}_2\text{O}$ in the revised manuscript;
- (2) “ $\text{Zn}(\text{NO}_3)_2$ ” in Figure 1 is revised as “ $\text{Zn}(\text{NO}_3)_2 \cdot 6\text{H}_2\text{O}$ ” in the revised manuscript;
- (3) “structural transformation from LIFM-CL1 to LIFM-CL1-H₂O” is revised as “structural transformation from LIFM-CL1-H₂O to LIFM-CL1” in Page 12;
- (4) “Zn-PCP” is revised as “Zn-MOF” in the caption of Supplementary Figure 6.

2. Page 7 Line 129, the authors say “reversible SC-SC structural transformation (Fig. 4d) without any luminescence loss after 10 cycles”, however, as shown in Fig. 4d, the luminescence intensities in vacuum gradually decrease, so the description is not quite exact. Moreover, as shown in Fig. 4d and Supplementary Figure 13, the powder and film of LIFM-CL1-H₂O show different quenching efficiency and different recovery time under the same conditions, why?

Response: Thank for the kind reminding. We have changed the statement to “reversible SC-SC structural transformation with slight luminescence loss after 10 cycles” instead of the previous expression.

As for the different quenching efficiency and different recovery time between the samples shown in Fig. 4d and Supplementary Figure 13, we believe they are caused by the different crystal morphology and size. In our recycling tests shown in Fig. 4d and Supplementary

Figure 13, one cycle is completed in such a way that the emission intensities have to reach the maximum and then recover to minimum. However, the powder sample in Figure 4d is microcrystals while the film sample in Supplementary Figure 13 is the ZnO-supported as-grown hybride crystalline film. The latter has much larger crystal size than the former. We have added the SEM images of two samples in Supplementary Figure 19. It is clear that the microcrystals used in Figure 4d show the size in the range of 1-5 μm (Supplementary Fig.19a), while the as-grown film crystals used in Supplementary Figure 13 show the size in the range of 10-15 μm (Supplementary Fig.19b). Since the time needed for the complete structural transition and luminescent change is dependent on the size of the crystals, different quenching efficiency and recovery time is observed for these two samples.

3. The authors present the Zn-MOF can remove water from solvents or dewater to solvents. In order to more clearly illustrate this point, the water content of solvents before and after being soaked with hydrated LIFM-CL1-H₂O and dehydrated LIFM-CL1 should be confirmed with other methods such as Karl Fischer titration or electrochemical methods. The water contents in fluorescence titration experiments also should be confirmed with other methods.

Response: Thank for the reviewer's suggestion. We have carried out the suggested supplementary experiments to confirm the processes including removing water from water containing solvent by the dehydrated Zn-MOF and dewatering to anhydrous solvent from the hydrated Zn-MOF. We selected the methanol solvent as an example and confirmed the water content under different conditions with the method of C30 Karl Fischer Coulometric Titrimetry (METTLER TOLEDO, Switzerland). To eliminate the influence from the residual water in the initial methanol solvent and experimental errors introduced during water addition, we firstly examined the water content of the initial dry methanol solvent (theoretically 0%) and the as-prepared methanol solvents by adding different amount of water (1.5-6.0 μL). And then, the water content of solvents before and after being soaked with hydrated LIFM-CL1-H₂O and dehydrated LIFM-CL1 samples were measured. The results are added in Supplementary Tables 4 and 5. It is evident that the dehydrated sample can adsorb water from the solvent containing water, while the hydrated sample can dewater to the dry solvent. As we respond above to the 5th question of Reviewer #2, the accurate and quantitative detection of traces of water in organic solvents depends on strict measurement conditions and establishment of precise analytic procedure, therefore, the water content in fluorescence titration was not performed.

Supplementary Table 4 | Examination of water content of the used CH₃OH solvents by Karl Fischer Coulometric Titrimetry.

Theoretical value	V (H ₂ O) / 3 mL CH ₃ OH	0	1.5 μL	3.0 μL	6.0 μL
	V / V	0%	0.05%	0.10%	0.20%
Measured value	Detected/ ppm (mg/kg)	316.4	957.3	1539.3	2866.1
	V / V	0.0251%	0.0758%	0.1219 %	0.2271 %

Supplementary Table 5 | Confirmation of water content of CH₃OH solvents before and after being soaked with hydrated LIFM-CL1-H₂O and dehydrated LIFM-CL1 by Karl Fischer Coulometric Titrimetry.

Mass of sample (mg)			hydrated	dehydrated
			3.1	2.2
Water content (v/v)	before	Measured/ppm (mg/kg)	316.4	2866.1
		V / V	0.0251%	0.2271 %
	after	Measured/ppm (mg/kg)	412.1	2718.5
		V / V	0.0326%	0.2154%

4. The single-crystal-to-single-crystal (SC-SC) transformation is carried out in accordance with the stoichiometric ratio, such as each Zn^{2+} needs one water, so I think the concentration of sensor might influence the detective results and the excess of sensor or water content might give different results. The author should conduct some experiments about it. Moreover, we don't know the water content of real samples, so how to choose the concentration of sensor?

Response: Thank for the reviewer's advice. The entire SC-SC structural transformation is indeed driven by the stoichiometric water molecules, i.e. one water per Zn^{2+} . However, as we discussed in the main text, such water capture and removal process can occur individually on a local Zn^{2+} site without influencing the whole crystal structure. In another word, the water sensing here can be accomplished just through a partial structural interconversion between the hydrated and dehydrated forms of the sample, and the emission change does not need the whole crystal conversion to obey exact stoichiometric ratio. For a qualitative detection of water molecules, the present Zn-MOF is competent for a convenient naked eye colorimetric sensor. To achieve accurate and quantitative determination of the water content, the exact stoichiometric ratio for the whole crystal SC-SC transformation may be necessary. In this paper, we have not intended to establish a precise analytical procedure for the quantitative detection of the water content. Further detailed study will be continued in due course.

REVIEWERS' COMMENTS:

Reviewer #2 (Remarks to the Author):

The authors have answered most of my questions. There are only a few minor corrections that should be considered:

1. Supplementary Figure 5. "(b) the PXRD pattern of LIFM-CL1-H₂O measured in ambient air (RH 98%)" should be "(b) the PXRD pattern of LIFM-CL1-H₂O measured in ambient air (RH 98%) compared with that of simulated pattern".

2. The reasons provided to explain the mixture phases shown in Supplementary Figure 12 should be incorporated either in the figure caption or main text so that the readers (not just the reviewer) understand why both phases co-exist. Also the authors don't seem to have addressed the question: "Likewise, the PXRD pattern after exposure to air should match the LIFM-CL1-H₂O, not LIFM-CL1."

Reviewer #3 (Remarks to the Author):

The authors have revised all the issues in detail. It is acceptable for publication in current form.

Reviewer #2 (Remarks to the Author):

The authors have answered most of my questions. There are only a few minor corrections that should be considered:

1. Supplementary Figure 5. “(b) the PXRD pattern of LIFM-CL1-H₂O measured in ambient air (RH 98%)” should be “(b) the PXRD pattern of LIFM-CL1-H₂O measured in ambient air (RH 98%) compared with that of simulated pattern”.

Response: Thank for reviewer’s kind reminder. The phrase has been modified as suggested.

2. The reasons provided to explain the mixture phases shown in Supplementary Figure 12 should be incorporated either in the figure caption or main text so that the readers (not just the reviewer) understand why both phases co-exist. Also the authors don’t seem to have addressed the question: “Likewise, the PXRD pattern after exposure to air should match the LIFM-CL1-H₂O, not LIFM-CL1.”

Response: Thank for reviewer’s kind suggestion. The reasons explaining the mixture phases have been incorporated into the figure caption of Supplementary Figure 12. And as for the question “Likewise, the PXRD pattern after exposure to air should match the LIFM-CL1-H₂O, not LIFM-CL1”, we are **very very** sorry that in the last revision, we forgot to change the error that “LIFM-CL1-H₂O and LIFM-CL1” are reversely positioned in Supplementary Figure 12, and now we have corrected the mistake.

Reviewer #3 (Remarks to the Author):

The authors have revised all the issues in detail. It is acceptable for publication in current form.

Response: Thank for reviewer’s positive appreciation.